# Inhibiting *Mycobacterium tuberculosis* CoaBC by targeting an allosteric site

Vitor Mendes [1✉], Simon R. Green[2], Joanna C. Evans[3], Jeannine Hess [4], Michal Blaszczyk[1], Christina Spry [4], Owain Bryant[1], James Cory-Wright[1], Daniel S-H. Chan[4], Pedro H. M. Torres[1], Zhe Wang[5], Navid Nahiyaan[5], Sandra O'Neill[2], Sebastian Damerow[2], John Post[2], Tracy Bayliss[2], Sasha L. Lynch[3], Anthony G. Coyne [4], Peter C. Ray[2], Chris Abell [4], Kyu Y. Rhee [5], Helena I. M. Boshoff [6], Clifton E. Barry III [3,6], Valerie Mizrahi[3], Paul G. Wyatt[2] & Tom L. Blundell [1✉]

Coenzyme A (CoA) is a fundamental co-factor for all life, involved in numerous metabolic pathways and cellular processes, and its biosynthetic pathway has raised substantial interest as a drug target against multiple pathogens including *Mycobacterium tuberculosis*. The biosynthesis of CoA is performed in five steps, with the second and third steps being catalysed in the vast majority of prokaryotes, including *M. tuberculosis*, by a single bifunctional protein, CoaBC. Depletion of CoaBC was found to be bactericidal in *M. tuberculosis*. Here we report the first structure of a full-length CoaBC, from the model organism *Mycobacterium smegmatis*, describe how it is organised as a dodecamer and regulated by CoA thioesters. A high-throughput biochemical screen focusing on CoaB identified two inhibitors with different chemical scaffolds. Hit expansion led to the discovery of potent and selective inhibitors of *M. tuberculosis* CoaB, which we show to bind to a cryptic allosteric site within CoaB.

[1] Department of Biochemistry, University of Cambridge, 80 Tennis Court Road, Cambridge CB2 1GA, UK. [2] Drug Discovery Unit, College of Life Sciences, University of Dundee, Dow Street, Dundee DD1 5EH Scotland, UK. [3] MRC/NHLS/UCT Molecular Mycobacteriology Research Unit & DST/NRF Centre of Excellence for Biomedical TB Research & Wellcome Centre for Infectious Diseases Research in Africa, Institute of Infectious Disease and Molecular Medicine and Department of Pathology, Faculty of Health Sciences, University of Cape Town, Anzio Road, Observatory 7925, Cape Town, South Africa. [4] Department of Chemistry, University of Cambridge, Lensfield Road, Cambridge CB2 1EW, UK. [5] Division of Infectious Diseases, Weill Department of Medicine, Weill Cornell Medical College, New York, NY 10065, USA. [6] Tuberculosis Research Section, Laboratory of Clinical Immunology and Microbiology, National Institute of Allergy and Infectious Disease, National Institutes of Health, 9000 Rockville Pike, Bethesda, MD 20892, USA. ✉email: vgm23@cam.ac.uk; tom@cryst.bioc.cam.ac.uk

Tuberculosis (TB) is the most prevalent and deadly infectious disease worldwide and remains a global epidemic. Despite the availability of treatment, this disease, caused by *Mycobacterium tuberculosis*, still claims 1.5 million lives each year[1]. Current treatment regimens are long, which presents an obstacle for patient adherence and imposes a heavy social and economic burden on countries with a high incidence of TB. It is therefore critical to explore novel targets and find new and more effective drugs to combat this disease.

Coenzyme A (CoA) is an essential and ubiquitous co-factor involved in numerous metabolic pathways with a large number of different enzymes requiring it for their activity[2]. CoA is essential for the synthesis of phospholipids, fatty acids, polyketides and non-ribosomal peptides, for the operation of the tricarboxylic acid cycle and in the degradation of lipids[3]. The importance of CoA for essential post-translational modifications of proteins is also well established in both eukaryotes and prokaryotes, with various proteins post-translationally modified by thioester derivatives of CoA (acylation) or CoA itself (phosphopantetheinylation and CoAlation), while several other post-translational modifications depend indirectly on CoA through the mevalonate pathway[4–7]. Furthermore, dephospho-CoA, an intermediate of the CoA pathway, is incorporated into some RNA transcripts during transcription initiation thereby serving as a non-canonical transcription initiating nucleotide[8]. These RNA modifications have functional consequences and occur in both eukaryotes and bacteria[8]. In *M. tuberculosis*, CoA plays a pivotal role in the biosynthesis of complex lipids that are crucial components of the cell wall and required for pathogenicity[9]. It is also needed for the degradation of lipids, including cholesterol, which are the primary source of energy for this organism during infection[10,11]. Given its ubiquitous nature, wide metabolic and functional impact of its inhibition, and lack of sequence conservation between prokaryotes and humans, the CoA pathway is therefore an attractive pathway for drug discovery for many different infectious diseases, including TB.

The biosynthesis of CoA from pantothenic acid (vitamin B₅) is performed in five steps (Fig. 1), sequentially catalysed by the enzymes pantothenate kinase (CoaA, also known as PanK), phosphopantothenoylcysteine synthetase (CoaB), phospho-pantothenoylcysteine decarboxylase (CoaC), phosphopantetheine adenylyltransferase (CoaD) and dephospho-CoA kinase (CoaE). However, in the vast majority of prokaryotes, including *M. tuberculosis*, CoaB and CoaC are encoded by a single gene to produce a fused bifunctional enzyme (CoaBC). Transcriptional silencing of individual genes of the CoA biosynthetic pathway of this pathogen identified CoaBC as uniquely bactericidal within the CoA pathway, highlighting it as a good candidate for drug discovery[12]. Nevertheless, the reported inhibitors for this enzyme are very few in number and almost invariably substrate mimicking[13–15].

CoaBC converts 4′-phosphopantothenate to 4′-phospho-pantetheine in three steps. First, 4′-phosphopantothenate (PPA) reacts with CTP to form 4′-phosphopantothenoyl-CMP with the release of pyrophosphate. This intermediate subsequently reacts with cysteine to form 4′-phosphopantothenoylcysteine (PPC) with the release of CMP, with these two steps being catalysed by CoaB. The product of CoaB is then decarboxylated by CoaC, an enzyme of the homo-oligomeric flavin-containing decarboxylase (HFCD) protein family, to 4′-phosphopantetheine. X-ray crystal structures have been reported for the individual CoaB and CoaC enzymes in several organisms, including a structure of CoaB from *Mycobacterium smegmatis*, a close relative of *M. tuberculosis*. However, a structure of a full-length bifunctional CoaBC had not been determined.

Here we report the structure of the bifunctional CoaBC of *M. smegmatis* at 2.5 Å. We identify a previously unknown allosteric site in CoaB and crucially, we report the discovery of the first *M. tuberculosis* CoaBC allosteric inhibitors. Using X-ray crystallography and enzyme kinetic experiments, we define the mode of binding of one of the inhibitors and show its impact on the protein structure and function. These results further illustrate the potential of CoaBC as a drug target in *M. tuberculosis*.

## Results

**Overall structure of CoaBC.** As the HFCD protein family of flavin-binding proteins are known to form homo-oligomers[16], we performed native electrospray-ionisation mass spectrometry (ESI-MS) to investigate the stoichiometry of CoaBC, previously proposed to form a dodecamer[16]. Both *M. tuberculosis* CoaBC (*Mtb*CoaBC) (Supplementary Fig. 1a) and *M. smegmatis* CoaBC

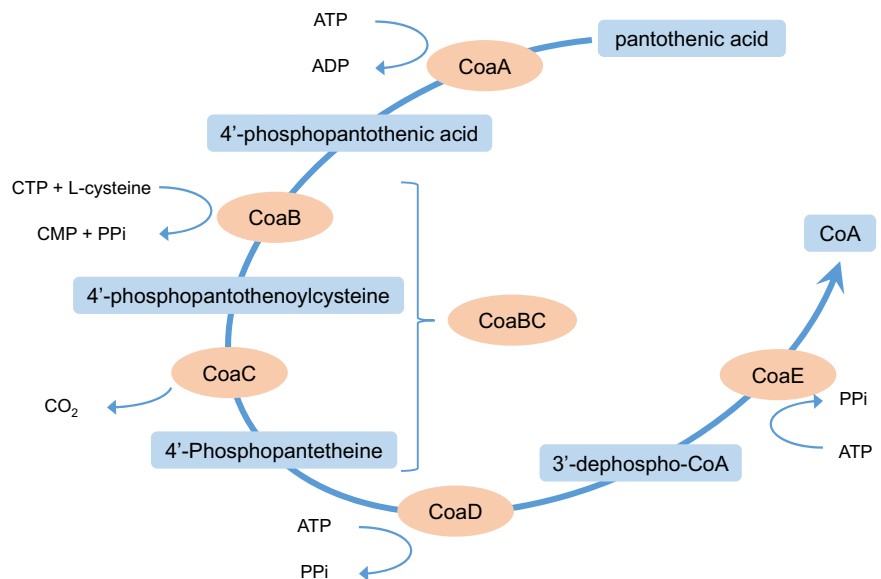

**Fig. 1 Coenzyme A biosynthesis pathway.** Depiction of the CoA biosynthesis pathway showing all the substrates and products in each biosynthetic step. Enzymes are circled in orange while pathway intermediates are boxed in blue.

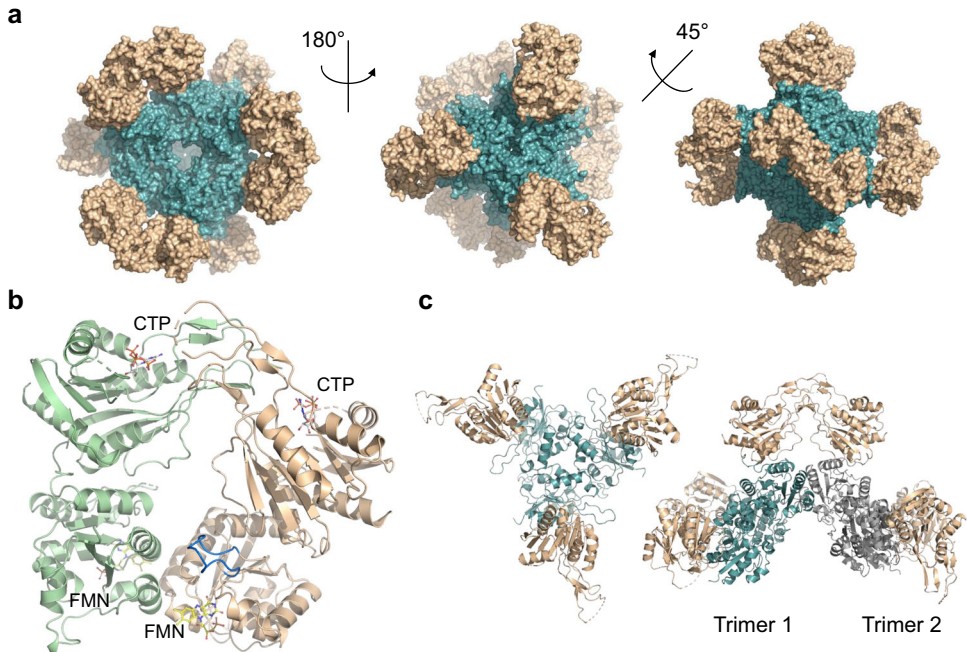

**Fig. 2 X-ray crystal structure of FMN and CTP-bound MsmCoaBC. a** Full aspect of the dodecameric CoaBC with CoaC represented in teal and CoaB in gold. **b** View of a CoaBC dimer with FMN and CTP shown. Each protomer is coloured differently. The CoaC active site flexible flap is highlighted in blue. **c** In the left panel, a CoaBC trimer is shown with the CoaC coloured in teal and CoaB in gold. On the right panel dimerisation of two CoaBC trimers is shown with CoaC coloured in teal or grey for different trimers. Each CoaB forms a dimer with protomers from different trimers.

(*Msm*CoaBC) (Supplementary Fig. 1b) exclusively exhibited a dodecameric assembly, with no other oligomeric species observed in the spectra, which is consistent with a strong interaction between the subunits of the complex. The dodecamer of *Mtb*CoaBC was centred around the 56+ charge state, with an observed mass of 537 kDa, while the dodecamer of *Msm*CoaBC was centred around the 52+ charge state, with an observed mass of 523 kDa. While the *Msm*CoaBC mass matches the expected mass including 12 bound flavin mononucleotide molecules, the *Mtb*CoaBC mass is ~1% higher than the expected mass of 530 kDa, which can be attributed to non-specific binding of solvent molecules or ions to the protein complexes under the soft ionisation conditions employed.

Structures of a few proteins of the HFCD family have been determined[17–21]. All of these structures show either a homo-trimeric or homo-dodecameric arrangement of the flavin-containing Rossmann fold with trimers forming at each of the vertices of the tetrahedron in the case of a dodecameric arrangement[18]. However, all of these proteins, unlike CoaBC, contain only a single functional domain. We solved the structure of *Msm*CoaBC (PDB: 6TGV) at 2.5 Å resolution (Fig. 2a), in the presence of CTP and FMN (Fig. 2b and Supplementary Fig. 2a, b), using crystals belonging to the H3₂ space group with an asymmetric unit containing four protomers forming two CoaBC dimers. Data collection and refinement statistics are summarised in (Supplementary Table 1). The final model (residues 2–412) covers both CoaC and CoaB, but densities for several residues in three loop regions in CoaB are not observed (residues 290–298, 336–342, 363–376). Nevertheless, all these residues except for 375 and 376, can be seen in the *Msm*CoaB X-ray crystal structure (PDB: 6TH2) that we also solved in this work at 1.8 Å. The *N*-terminal CoaC of *Msm*CoaBC (residues 1–179) forms the same type of dodecameric arrangement seen in other HFCD family proteins, such as the peptidyl-cysteine decarboxylase EpiD[18], and it sits at the core of the dodecamer (Fig. 2a, c) with the two domains connected through a small loop region (residues

180–189) that tightly interacts with both. The active site of CoaC sits at the interface between two protomers of one CoaC trimer and a protomer of an adjacent CoaC trimer with the FMN site facing inwards towards the hollow centre of the dodecamer (Fig. 3a). A previously described flexible flap that encloses the reaction intermediate bound to *Arabidopsis thaliana* CoaC[22] is also observed in some of the protomers, but in an open conformation (Fig. 2b).

The *C*-terminal CoaB of *Msm*CoaBC also displays a Rossmann fold consistent with several other CoaB structures solved previously, including both the eukaryotic form, in which CoaB exists as an individual polypeptide, and the bacterial form where CoaB is typically fused with CoaC[23–25]. Each CoaB of *Msm*CoaBC (residues 190–414) dimerises with a CoaB belonging to an adjacent trimer (Fig. 2c). The full protein resembles a tetrahedron with CoaB dimers positioned at the six edges and CoaC trimers at the four vertices (Fig. 2a).

The shortest distance between a pair of CoaB and CoaC active sites is ~30 Å (Fig. 3b). Nevertheless, a flexible loop (residues 362–377) that covers the 4′-phosphopantothenate site, when this substrate binds to the enzyme[23], can be seen in our *Msm*CoaB structure, extending away from the active site. A superposition of our *Msm*CoaB dimer structure with *Msm*CoaBC shows the loop extending towards the CoaC active site (Fig. 3b). This long loop (15–16 amino acids) is present in all CoaBCs (Supplementary Fig. 3) and it is possible that it helps channelling the substrate from the CoaB to the CoaC active site.

The small differences (RMSD = 1.147 Å) in overall structure of CoaB dimers in the full-length *Msm*CoaBC and the *Msm*CoaB crystal structure solved at 1.8 Å (Supplementary Fig. 4) can be attributed to artefacts of crystal packing. Similarly, the CoaC structure does not seem to differ between full-length *Msm*CoaBC and the available individual CoaC structures. However, when *Msm*CoaB (residues 186–414) is expressed alone, the protein does not dimerise in solution and is inactive. This contrasts with *E. coli* CoaB, which still dimerises and is functional when expressed on its own without

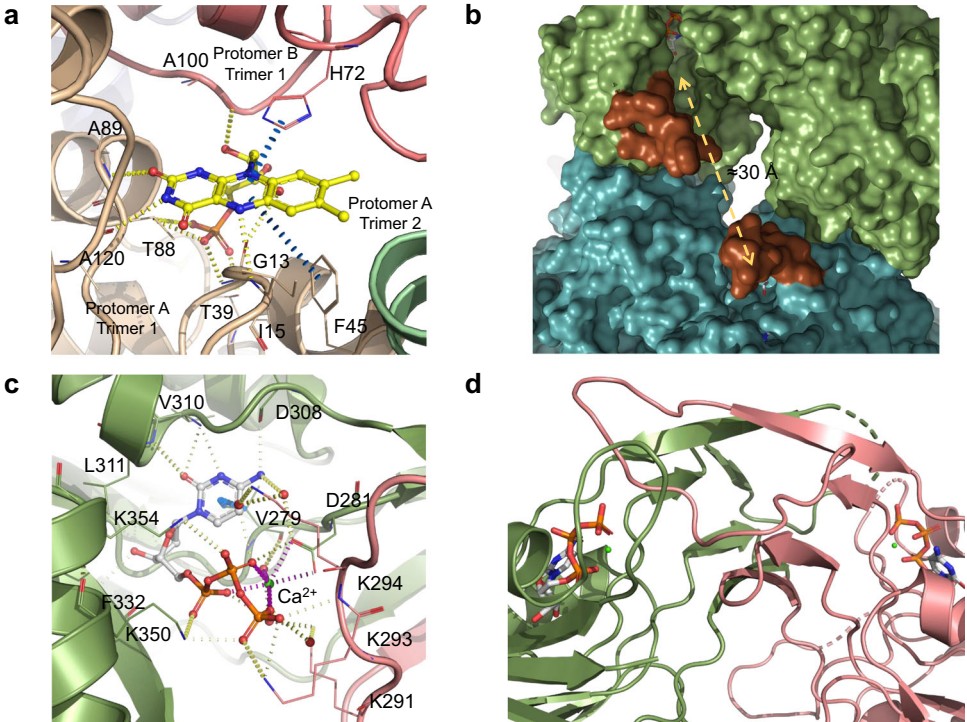

**Fig. 3 Detailed view of MsmCoaBC active sites and MsmCoaB dimerisation interface. a** View of CoaC active site with FMN bound. The active site sits between two protomers of one trimer (gold and pink) and a third protomer from an adjacent trimer (green). Hydrogen bonds are depicted in yellow and π-interactions are in blue. **b** Superposition of a CoaB crystal structure in green, with full-length CoaBC (teal) showing the active site flaps (brown) of the CoaB and CoaC enzymes. **c** Detailed view of the CTP binding site. Cartoon and residues belonging to each protomer are coloured differently. Hydrogen bonds and π-interactions are coloured as in **b**. Important waters are represented as red spheres and calcium as a green sphere. Calcium coordination is depicted in purple. **d** CoaB dimerisation interface. Each protomer is coloured as in **c**.

the N-terminal CoaC[13,26]. The CoaB dimer interface is mostly conserved, but there are clear differences in the dimerisation region between MsmCoaB and E. coli CoaB that could help to explain the different observed oligomerisation patterns (Supplementary Fig. 3). The absence of dimerisation for the MsmCoaB when expressed alone suggests that the interactions between CoaC and CoaB in M. smegmatis, and likely all other Mycobacteriaceae, are fundamental for CoaB dimerisation and activity. This idea is reinforced by the fact that the residues located at the interface of the two enzymes (CoaB and CoaC) are well conserved in all Mycobacteriaceae and somewhat conserved in the sub-order Corynebacterineae, but not outside of this group (Supplementary Fig. 3).

The CoaB dimerisation region forms a β-sandwich composed of eight anti-parallel β-strands, related by 2-fold symmetry, that contacts with the active site (Fig. 3c, d). Comparison of the MsmCoaB with human CoaB reveals that the human and many other eukaryotic CoaBs[24] possess two extra α-helices and β-strands involved in the dimerisation interface that help stabilise the dimer in the absence of CoaC (Supplementary Fig. 5).

The CoaB active site is enclosed by a loop that extends from the opposing protomer. This loop contains a motif "K-X-K-K", which is widely conserved in bacteria (Supplementary Fig. 3), with few exceptions, and each lysine either interacts directly with the triphosphate group of CTP or through highly coordinated waters (Fig. 3c). Also observed is the coordination of a cation by the triphosphate group and D281 (Fig. 3c and Supplementary Fig. 6). While magnesium or manganese are the favoured cations for CoaB activity[27], calcium is observed in our structures instead, due to the high concentration present in the crystallisation condition.

**CoaBC is regulated by CoA thioesters**. It is known that CoA biosynthesis is regulated, in many organisms, at the first step of

the pathway, which is catalysed by the enzyme CoaA[3]. M. tuberculosis and many other mycobacteria possess a CoaA (type I PanK) as well as CoaX (type III PanK). However, only the type I PanK seems to be active based on studies in M. tuberculosis[28]. CoA and its thioesters competitively inhibit E. coli CoaA by binding to the ATP site, with CoA being the strongest regulator[29,30]. Nevertheless, at physiologically relevant levels of CoA there is only a low level inhibition of CoaA[30]. It is also known that M. tuberculosis CoaD, the enzyme that catalyses the fourth step of the pathway, is competitively inhibited by CoA and its product dephospho-CoA[31,32]. However, nothing was known about the regulation of CoaBC in any organism. We therefore examined the effect of CoA and several of its thioesters (acetyl-CoA, malonyl-CoA and succinyl-CoA) on MtbCoaBC activity, using a coupled enzymatic assay that quantifies the release of pyrophosphate (EnzChek pyrophosphate assay). Controls were performed to assess the activity of these compounds against the two coupling enzymes and the compounds showed an absence of inhibition at the tested range of concentrations.

Inhibition of CoaB activity by CoA and acyl-CoAs was observed, with $IC_{50}$ values ranging from 38 to 148 μM, far below the predicted intracellular concentrations of acyl-CoAs[33], with succinyl-CoA displaying the highest inhibition (Fig. 4a and Table 1). To assess the mode of inhibition of CoA and its thioesters the EnzChek-coupled enzyme assay that measures the release of pyrophosphate was used to follow the first half of the reaction while a CMP quantification assay was used for the second half of the reaction. Competition assays with the three substrates and acetyl-CoA show a competitive mode of inhibition relative to CTP and PPA with a $K_i$ of 22.5 and 22.4 μM, respectively, and non-competitive inhibition for L-cysteine with a $K_i$ of 62.1 μM (Fig. 4b and Table 2). In the absence of a crystal

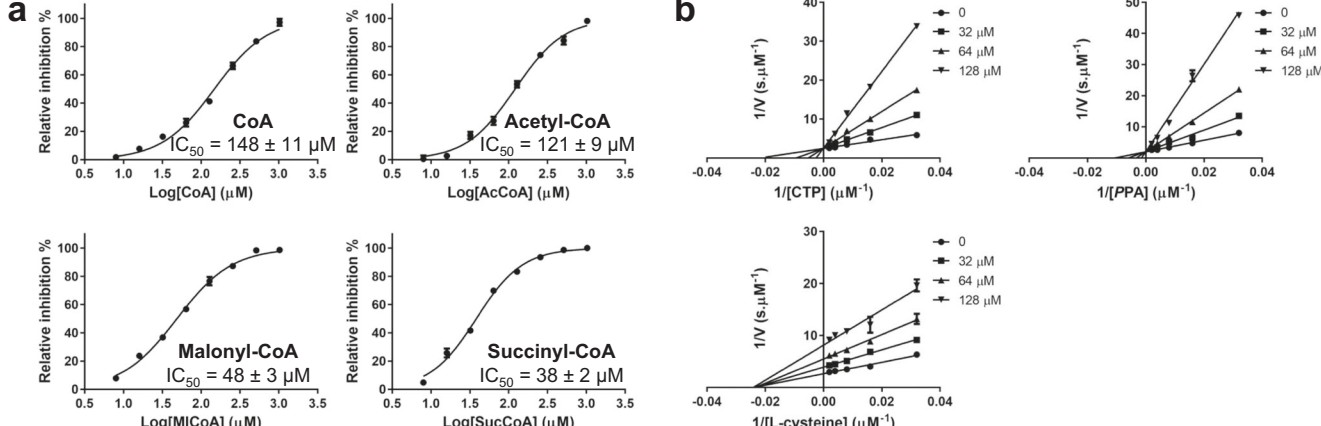

**Fig. 4 Regulation of MtbCoaBC by CoA and CoA thioesters. a** Inhibition of *Mtb*CoaBC by CoA, acetyl-CoA, malonyl-CoA and succinyl-CoA.
**b** Lineweaver–Burk plots showing the effect of varying the concentration of each substrate in the presence of different concentrations of acetyl-CoA. Data are presented as average values of three independent experiments with ± SD. Source data are provided as a source data file.

**Table 1 Inhibition of *Mtb*CoaB domain by CoA, CoA thioesters and *Mtb*CoaB, *Msm*CoaB and HCoaB by the most potent inhibitors from of series one and two.**

| Compound | IC$_{50}$ EnzChek *Mtb*CoaBC (µM) | IC$_{50}$ EnzChek HCoaB (µM) | IC$_{50}$ EnzChek *Msm*CoaBC (µM) |
|---|---|---|---|
| CoA | 148 ± 11 | | |
| AcCoA | 121 ± 9 | | |
| MlCoA | 49 ± 3 | | |
| SucCoA | 38 ± 2 | | |
| 1b | 0.28 ± 0.05 | 122 ± 22 | 0.26 ± 0.02 |
| 1c | 4.6 ± 0.4 | | |
| 2b | 0.08 ± 0.01 | 60 ± 7 | 0.18 ± 0.02 |
| 2c | 0.41 ± 0.03 | | |
| 2d | 0.54 ± 0.06 | | |
| 2e | 3.0 ± 0.2 | | |

IC$_{50}$ values determined using the EnzChek pyrophosphate assay are shown.
Data are presented as average values of three independent experiments with ± SD.

**Table 2 Inhibitor constants of acetyl-CoA, compounds 1b and 2b for the three CoaB substrates.**

| Inhibitor | Variable substrate | $K_i$ (µM) | Inibition type# |
|---|---|---|---|
| AcCoA | CTP | 22.5 ± 1.7 | C |
| | PPA | 22.4 ± 1.4 | C |
| | L-cysteine | 62.1 ± 2.0 | NC |
| 1b | CTP | *0.222 ± 0.012 | UC |
| | PPA | *0.078 ± 0.005 | UC |
| | L-cysteine | *0.181 ± 0.008 | UC |
| 2b | CTP | 0.093 ± 0.018 | Mixed |
| | PPA | *0.062 ± 0.004 | UC |
| | L-cysteine | *0.055 ± 0.003 | UC |

*For uncompetitive inhibitors α$K_i$ product is shown.
#Abbreviations: *C* competitive inhibition, *NC* non-competitive inhibition, *UC* uncompetitive inhibition.
Data are presented as average values of three independent experiments with ± SD.

structure to confirm the mode of binding, these results suggest that acyl-CoAs most likely bind to the active site itself, competing directly with CTP and PPA. Interestingly, both acyl-CoAs, involved in fatty acid synthesis, as well as those involved in the

TCA cycle, show inhibition of CoaB, with larger fatty acyl chains showing higher inhibition of CoaB (Fig. 4a).

**Identification of CoaB inhibitors using high-throughput screening.** Although the CoA biosynthetic pathway is considered an attractive target for drug discovery, CoA pathway inhibitors displaying potent whole cell activity are rare and the few CoaBC inhibitors that have been reported to date are in majority substrate mimicking[13,34].

In order to identify *Mtb*CoaBC inhibitors, we have conducted a high-throughput screen of 215,000 small molecules targeting CoaB activity. To do this, an end-point pyrophosphate quantification assay was used (Biomol Green). The most potent hits identified were compounds 1a and 2a with IC$_{50}$ values of 9 and 3.1 µM, respectively (Fig. 5 and Supplementary Table 2), originating from two different but related chemical scaffolds. A search was then performed for commercially available analogues. Testing of analogues of the initial hits resulted in the identification of more potent compounds with sub-micromolar IC$_{50}$ values (Table 1 and Supplementary Table 2, Figs. 5, 6a and Supplementary Fig. 7). Of these, compounds 1b and 2b (Figs. 5, 6a and Table 1), with IC$_{50}$ values of 0.28 and 0.08 µM, respectively, were identified as the most potent of the two chemical series and therefore were selected for further work.

**Selectivity and elucidation of the mode of inhibition.** Following the identification of potent *Mtb*CoaB inhibitors we went on to determine their activity against human CoaB (HCoaB) and *Msm*CoaBC. Control experiments were first performed to assess compound activity against the two coupling enzymes of the EnzChek assay and the compounds were found to be inactive at 100 µM. The IC$_{50}$ values for the most active compounds against *Mtb*CoaB were re-determined with this assay and the values obtained were in line with the primary screening assay (Supplementary Table 2). Both compounds were also found to be highly selective for mycobacterial enzymes showing activity for *Msm*CoaBC in a similar range to what was observed for *Mtb*CoaBC with IC$_{50}$ values of 0.26 and 0.18 µM, respectively, and were two orders of magnitude less active for the human CoaB (HCoaB) with IC$_{50}$ values of 122 and 60.4 µM, respectively (Table 1 and Supplementary Fig. 7).

Competition experiments were then performed between the three CoaB substrates and the two most potent compounds of each chemical series (1b and 2b) to determine the mode of inhibition. Compound 1b showed uncompetitive inhibition for all

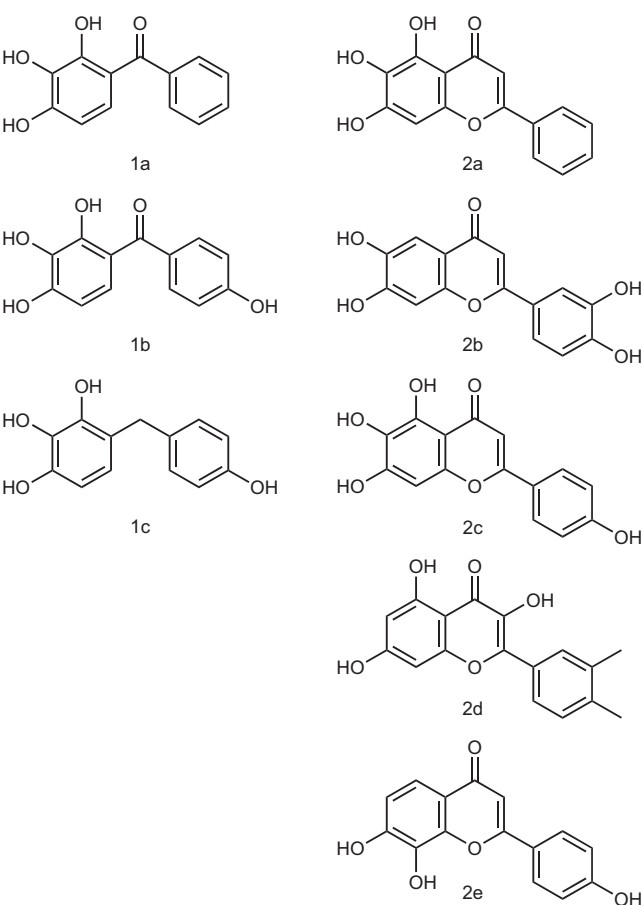

**Fig. 5 Chemical structures of the main compounds of series 1 and 2.**
Chemical structures of the initial hits (1a and 2a) and most potent
compounds for series 1 and 2.

substrates with a $\alpha K_i$ of 0.222, 0.078 and 0.181 μM, respectively,
for CTP, PPA and L-cysteine (Fig. 6b and Table 2), consistent
with the compound binding preferentially when the three
substrates are bound. Compound 2b shows mixed inhibition
relative to CTP with a $K_i$ of 0.093 μM and uncompetitive
inhibition for PPA and L-cysteine with a $\alpha K_i$, respectively, of
0.062 and 0.055 μM (Fig. 6c and Table 2). It is known that CoaB
forms the phosphopantothenoyl-CMP intermediate in the
absence of L-cysteine[35] and, due to spatial constraints, it is likely
that cysteine can only bind at the active site after the release of
pyrophosphate. The data is therefore consistent with compound
1b preferentially binding after L-cysteine enters the active site, for
the last step of catalysis and the formation of 4′-phosphopan-
tothenoylcysteine and CMP. However, compound 2b shows a
mixed inhibition for CTP, reflecting a slightly different mechan-
ism of action. These results obtained for both compounds suggest
the existence of an allosteric site in the CoaB moiety of
*Mtb*CoaBC.

**Structural basis for inhibition of CoaB by allosteric inhibitors.**
In order to elucidate the binding mode of compound 1b we used
a truncation of the *Msm*CoaB (residues 187–414) that was pre-
viously crystallised before by others in the presence of CTP (PDB:
4QJI) at 2.65 Å resolution. Screening for crystallisation conditions
allowed us to find a different CTP containing condition that gave
excellent resolution (1.8 Å). Comparison of this structure with the
full-length *Msm*CoaBC (Supplementary Fig. 4) showed only
minor differences that can be attributed to crystal packing. Hence

this crystallisation system could be used to validate CoaB inhi-
bitors binding outside of the CTP site.

*Msm*CoaB was co-crystallised with CTP in the absence of
compound 1b and overnight soaking of the crystals with this
compound was performed. A co-crystal structure of *Msm*CoaB
with compound 1b (PDB: 6THC) was obtained and showed that
the compound was bound to a site at the dimer interface of CoaB,
in a deep cavity that is occluded when the compound is absent
(Fig. 7a, b and Supplementary Fig. 8). Each CoaB dimer contains
two of these sites, which are formed by a sandwich of eight β-
strands and a long loop that contains the conserved "K-X-K-K"
motif. This site opens to the active site and the inhibitor also
contacts with D281 that is involved in the coordination of the
cation (Fig. 7c). Several residues are shared between the allosteric
site and the active site and these include R207, N211, A280, D281
I292, K293 and K294. The opening/closing of this cryptic
allosteric site is mediated by the side chain of R207 of the
opposing protomer (Fig. 7d) that moves 5.5 Å at the furthest
point and, to a smaller extent, by the side chain of F282 that
moves 2 Å. R207 has previously been shown to be critical for the
second half of the reaction catalysed by CoaB, the conversion of
the 4′-phosphopantothenoyl-CMP intermediate to PPC, with
almost no conversion of the intermediate to PPC detected when
this arginine is mutated to glutamine[35]. Given the position of this
arginine, it is likely that it is involved in the binding of cysteine.
Despite the absence of a crystal structure with cysteine, kinetic
data showing uncompetitive inhibition with cysteine is consistent
with this.

This allosteric site is comprised of a large group of
hydrophobic residues (I209, F282 and L304 of protomer A and
L203, I292, P299 and I302 of protomer B) many of which form
hydrophobic interactions with compound 1b (Fig. 7c). Several π-
interactions between the compound and the protein are also
observed and involve the backbone of D281 and the side chain of
F282 of protomer A and R207 of protomer B (Fig. 7d). Hydrogen
bond interactions are formed with D281 and F282 of protomer A
and R207 of protomer B. Water-mediated interactions are also
observed for a group of residues that sit at the outer edge of the
site (L203, H286 and D303) that is formed exclusively by
protomer B (Fig. 7c).

We propose that upon binding of L-cysteine, the R207 side chain
moves towards the active site, and is likely involved in stabilising/
orienting L-cysteine to attack the phosphopantothenoyl-CMP
intermediate. This movement opens the allosteric site, which allows
binding of allosteric inhibitors to the newly created cavity. The
allosteric inhibitors will then stabilise the enzyme in its substrate-
bound state with the position of R207 becoming locked by several
hydrogen bonds with the side chain of D281 of protomer A, the
backbone carbonyl group of I292 and the side chain of D204 of
protomer B, but also by the π-interactions with the compound
(Fig. 7c). The residues around this site and crucially R207 are
conserved across many microorganisms, suggesting that this
allosteric site is present in most, if not all bacterial CoaBCs
(Supplementary Figs. 3, 9a). Interestingly, even though overall
sequence identity is very low between the human CoaB and
*Msm*CoaB (22%), the human enzyme also contains an arginine
equivalent to R207 and a roughly similar interface with several
conserved residues, but there are stark differences in the relative
position of the residues at this site between the two enzymes
(Supplementary Fig. 9b) which can explain the observed large
differences in potency of the inhibitors between the human and
mycobacterial enzymes.

While we were not able to obtain co-crystal structures with
other inhibitors, in silico docking helped to provide a possible
explanation for the structure–activity relationship observed for
series one and two. The highest-scoring docking pose of

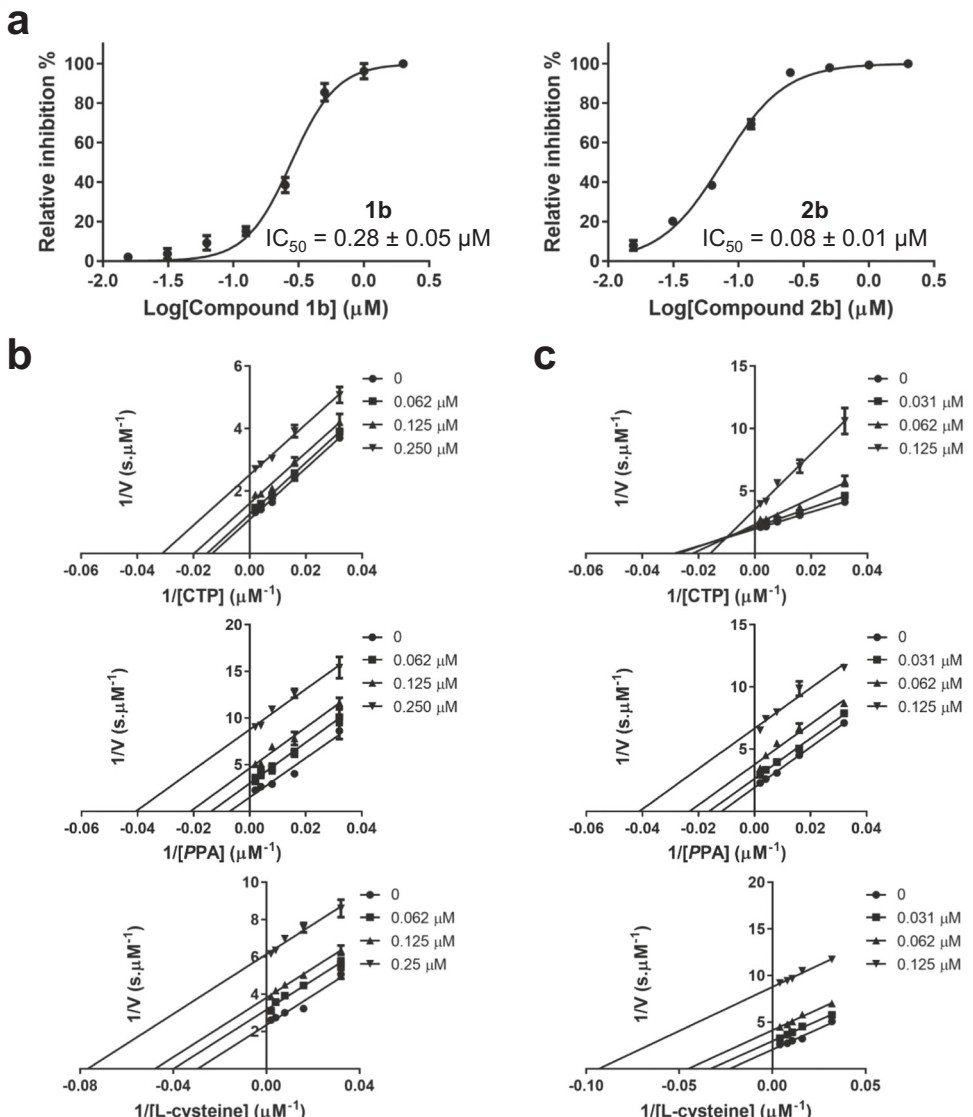

**Fig. 6 Inhibition of MtbCoaBC by compounds 1b and 2b. a** Dose response profiles and chemical structure of compounds 1b and 2b is shown.
**b, c** Lineweaver–Burk plots respectively showing the effect of varying concentrations of compounds 1b and 2b in the presence of varying concentrations of CTP, PPA and L-cysteine. Data are presented as average values of three independent experiments with ± SD.

compound 1b, the most potent inhibitor of series one, was almost identical to that observed in the co-crystal structure (Supplementary Fig. 10a), and the analogues for which docking was performed adopted a similar binding pose. The lower activity of compound 1a relative to compound 1b could be explained by the loss of water-mediated hydrogen bonds (Fig. 7c and Supplementary Fig. 10b), while the lower activity of compound 1c could be explained by the loss of the carbonyl group which faces a highly electropositive area of the protein (Supplementary Fig. 10c). Compound 2b is predicted to form direct hydrogen bonds at the bottom of the allosteric site, similar to those formed by compound 1b, but also to interact directly with L203 and H286, forming extra hydrogen bonds at the top of the allosteric site (Fig. 8). For compound 1b the interactions at the top of the site are water mediated (Fig. 7c). This could explain the higher potency of compound 2b. Compounds 2c and 2d are also predicted to form direct hydrogen bond interactions at the top of the allosteric site, but the interactions at the bottom of the site are not as favourable due to the presence of extra hydroxyl groups (Supplementary Fig. 10d–f). The remaining compounds in series

two, which have fewer hydroxyl groups and/or hydroxyl groups in different positions, lose the ability to form hydrogen bonds, consistent with the weaker inhibitory effect observed (Supplementary Table 2). The observed loss of activity of these compounds demonstrates the importance of the catechol group for binding in these two chemical series.

**Screening of CoaBC inhibitors against *M. tuberculosis*.** The in vitro whole cell activity of the compounds was further evaluated by their ability to inhibit *M. tuberculosis* growth on different carbon sources. None of the compounds exhibited activity in media containing glycerol or cholesterol as the main carbon source (Table 3). We then tested whether the lack of inhibitory activity could be attributed to the presence of BSA by determining the whole cell activity of the three most potent inhibitors against *M. tuberculosis* in GAST/Fe minimal media. All the tested compounds exhibited moderate to low activity in this media with compound 2b displaying the best activity of the five (Table 3). The compounds were then tested against *M. tuberculosis* using a

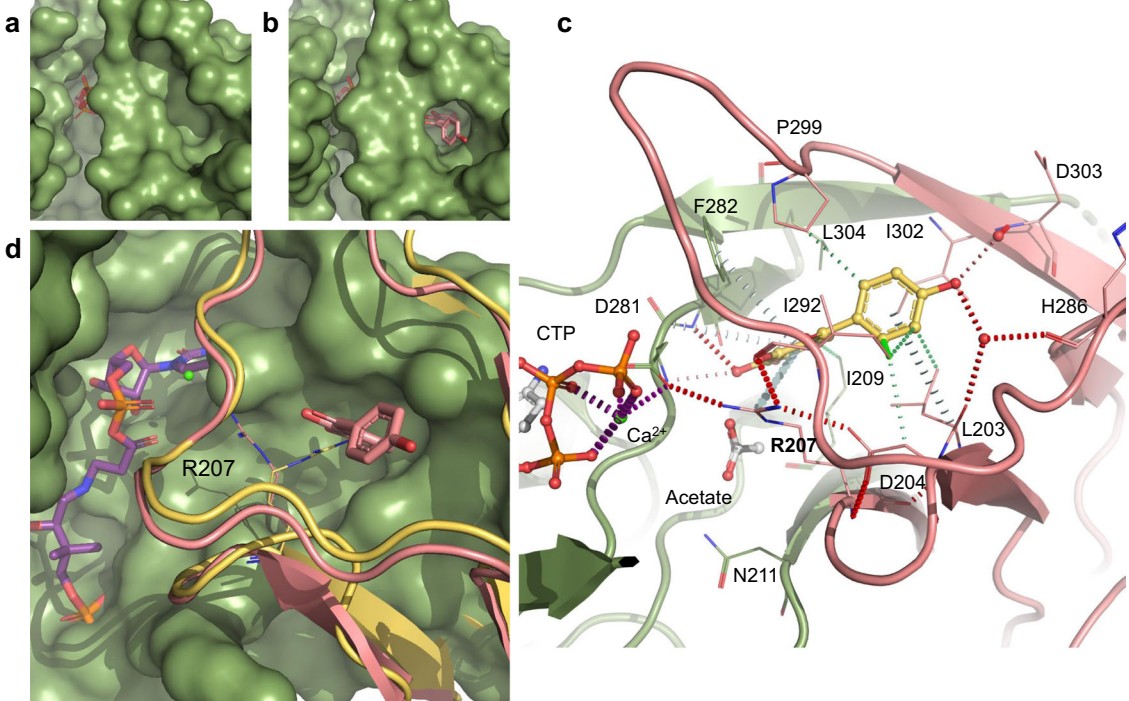

**Fig. 7 MsmCoaB X-ray structure showing the cryptic allosteric site.** CoaB with the cryptic allosteric site closed (**a**) and opened conformation (**b**) with compound 1b (pink) bound. **c** Detailed view of the allosteric site with compound 1b (yellow) bound. The individual protomers of the CoaB dimer are coloured in green or pink. Hydrogen bonds are depicted in red, π-interactions are in grey and hydrophobic interaction in green. Important waters are represented as red spheres and calcium as a green sphere. Calcium coordination is depicted in purple. **d** Gating mechanism of the cryptic allosteric site showing the movement of R207 with the closed conformation in yellow and the open conformation in pink. An *E. coli* structure (PDB: 1U7Z) with the 4'-phosphopantothenoyl-CMP (purple) intermediate bound is superimposed.

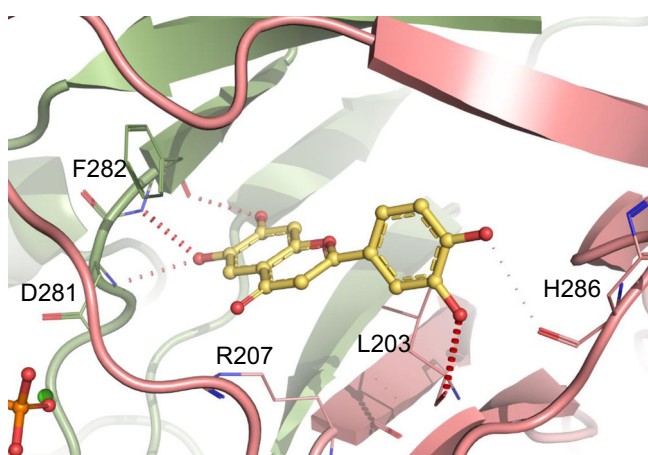

**Fig. 8 Docking of compound 2b into MsmCoaB showing the highest-scoring pose.** Hydrogen bonds are shown in red. The individual protomers of the CoaB dimer are either coloured in green or pink.

**Table 3 Minimum inhibitory concentration (MIC) values of CoaB inhibitors against *M. tuberculosis* H37Rv cultured in different media (μM).**

| Compound | 7H9/ADC/Glycerol | 7H9/Cholesterol/Tyloxapol | GAST/Fe |
|---|---|---|---|
| 1b | >250 | >250 | 125 |
| 1c | >250 | ND | 125 |
| 2b | >250 | >250 | 50 |
| 2c | >250 | >250 | 125 |
| 2d | >250 | >250 | >250 |
| 2e | >250 | ND | 125 |

*ND* Not determined.
Data are presented as average values of three independent experiments.

CoaBC inducible knockdown[12] in GAST/Fe minimal media to determine whether the observed anti-mycobacterial activity was related to CoaBC inhibition. However, incremental silencing of CoaBC did not sensitise *M. tuberculosis* further to compounds 1b and 2b, suggesting that the activity of the compounds is not on target. We further tested if the compounds could permeate *M. tuberculosis* cell envelope and reach the cytoplasm. Compound 1b exhibited no depletion from the media indicating a lack of uptake or rapid efflux (Supplementary Table 3). Compound 2b, in contrast, exhibited complete depletion from the media but accumulated in *M. tuberculosis* to levels that accounted for less than 2%

of the amount consumed (Supplementary Table 3), indicative of its likely intrabacterial metabolism[36–39]. These results explain the observed differences in potency between the enzymatic assay and whole cell activity and shows that they are related to low compound permeation/high efflux for 1b and metabolism in the case of 2b.

## Discussion

CoA is an essential co-factor ubiquitous across all domains of life. For many years, this pathway has been considered an attractive drug target to develop new antibiotics against a wide range of pathogens including *M. tuberculosis*[40]. Furthermore, the recent identification of CoaBC as a key fragility point in the CoA pathway of this organism[12], combined with the low sequence

identity with the human CoaB (25%), makes this enzyme a highly attractive drug discovery target.

While a structure of an individual mycobacterial CoaB was available, we were aware that the many questions remaining at the start of this work about the organisation and regulation of this bifunctional enzyme could have significant implications for drug discovery. We therefore set out to obtain a full-length structure of a mycobacterial CoaBC and we successfully solved the *Msm*CoaBC structure, which shares very high sequence identity with the *M. tuberculosis* orthologue (86% full-length, 84% CoaB enzyme) and hence is a valuable tool for studying *M. tuberculosis* CoaBC. The organisation of CoaBC is similar to other HFCD family proteins[18] but unique in the sense that it contains more than one domain and highlights how the arrangement of the fused enzymes is essential for mycobacterial CoaB dimerisation and function. This fused arrangement might also help to channel the CoaB product to the CoaC active site more effectively. The human CoaB and other eukaryotic orthologues form stable dimers due to the extra dimerisation region (Supplementary Fig. 5), but it is also known in yeast that the entire CoA pathway assembles into a metabolon centred on CoaC[41,42] (known as CAB3). This hints that close proximity between the different active sites of the CoA enzymes is desirable and that substrate channelling of products and substrates between different enzymes might be important in this pathway. It is not clear at this point if such an arrangement for the entire CoA pathway is also present in bacteria.

Regulation of the CoA biosynthesis pathway was known to occur for other enzymes of the pathway through feedback inhibition by CoA, but no information was available for CoaBC. We demonstrate that both CoA, as well as several CoA thioesters, regulate CoaBC by inhibiting CoaB activity, and that these molecules act competitively for CTP and PPA and non-competitively for L-cysteine. This is consistent with these molecules binding to the CoaB active site but not to the L-cysteine sub-site. CoA and acyl-CoAs inhibit both CoaA and CoaD enzymes to varying extents[30,32,43]. However, the inhibitory effect of CoA and its thioesters on the activity of these enzymes is lower when compared to what we observed in CoaBC and consequently the impact of the intracellular level of these molecules will be predominant in CoaBC. We therefore report a new and important mechanism of regulation of "de novo" CoA biosynthesis, mediated by the action of CoA thioesters on CoaBC. Since the reported intracellular levels of these molecules[33] are normally above the observed $IC_{50}$, the activity of CoaBC is highly inhibited. This correlates well with previous work showing that "de novo" CoA biosynthesis closely matches dilution due to cell division[44]. However, the data for intracellular concentrations of CoA and CoA thioesters, as well as CoA half-life was not obtained for mycobacteria, and both interspecies differences along with variations in growth conditions may affect these conclusions.

Although the CoA pathway and CoaBC have been the subject of many drug discovery efforts, few non-substrate-mimicking inhibitors of CoaBC have been reported[13]. Our work identifies two related chemical scaffolds that potently inhibit the activity of the CoaB moiety of *Mtb*CoaBC through a cryptic allosteric site that sits in the dimer interface region of the CoaB enzyme. This site is closed in the CTP-bound structure, by the side chain of R207 a residue known to be involved in the second and final step of the reaction catalysed by CoaB – the conversion of the 4'-phosphopantothenoyl-CMP intermediate to PPC[35]. Considering the role of this residue in the final step of product formation and that compound 1b shows uncompetitive inhibition relative to all CoaB substrates, we propose that the opening of this site occurs upon binding of the final substrate L-cysteine. Currently it is not clear whether this allosteric site is exploited by a natural ligand, as

we were unable to identify such a biomolecule. Nevertheless, the conservation of residues at this site, across a variety of bacteria, indicates that this feature might be common to many, if not all, bacterial CoaBs.

The different mechanism of inhibition between compounds 1b and 2b in relation to CTP is difficult to explain without a structure of a complex between CoaB and 2b and docking does not offer further insight. However the observed mixed inhibition model for CTP in the case of compound 2b might be due to the fact that this compound is more sterically bulky than 1b and slight movements of residues at the bottom of the allosteric site (A280 and D281) are required to accommodate the compound and that these movements are blocked in the crystal. This correlates well with the fact that no crystal structure of compounds of series 2 could be obtained even when soaking at high concentrations.

Drug discovery against *M. tuberculosis* is rich in examples of compounds with potent activity against an essential enzyme but with a complete lack of whole cell activity due to the impermeable cell wall of this organism, efflux pumps, target modification enzymes and extensive capacity to metabolise compounds[45]. The modest in vitro whole cell activity displayed by the CoaB inhibitors reported in this work is off target and relates to the characteristics of *M. tuberculosis* with compound 1b being unable to reach the cytoplasm and 2b suffering metabolism. Nevertheless, the biochemical and structural data described herein further validate CoaBC as a promising anti-tubercular drug target by showing an allosteric site that can be targeted by potent and selective inhibitors.

## Methods

**Cloning and protein purification.** *M. tuberculosis* and *M. smegmatis coaBC* genes were amplified from genomic DNA of *M. tuberculosis* H37Rv strain, obtained from ATCC (ATCC25618D-2), using the primers *Mtb*BC28S-F and *Mtb*BC28S-R (Supplementary Table 4), and genomic DNA of *M. smegmatis* mc² 155, using the primers *Msm*BC28S-F and *Msm*BC28S-R (Supplementary Table 4), and cloned into a pET28a vector (Novagen), modified to include an *N*-terminal 6xHis-SUMO tag. The *M. smegmatis coaB* construct was obtained from the Seattle Structure Genomics Center for Infectious Disease. An *E. coli* optimised Human CoaB gene was ordered from Thermofisher and cloned into pHAT4[46] using NcoI and HindIII restriction sites. The same protein purification protocol was used for both *M. tuberculosis* and *M. smegmatis* CoaBC constructs.

*E. coli* BL21(DE3) containing pET28aSUMO-CoaBC was grown in 2XYT media at 37 °C until an $O.D._{600} = 0.6$. IPTG was then added to a final concentration of 0.5 mM and the temperature changed to 18 °C for 18–20 h. Cells were then harvested by centrifugation, re-suspended in 50 mM TRIS pH 8.0, 250 mM NaCl, 20% (w/v) glycerol, 20 mM imidazole, 5 mM $MgCl_2$ with protease inhibitors tablets (Roche) and DNAseI (Sigma). Cells were lysed with an Emulsiflex (Avestin) and the resultant cell lysate was centrifuged at 27,000× *g* for 30 min to remove cell debris. Recombinant CoaBCs were purified with a HiTrap IMAC Sepharose FF column (GE-Healthcare), equilibrated with 50 mM TRIS pH 8.0, 250 mM NaCl, 20% (w/v) glycerol and 20 mM imidazole. Elution was performed in the same buffer with 500 mM imidazole. Protein was dialysed in 25 mM TRIS pH 8 and 150 mM NaCl and the SUMO tag was cleaved overnight at 4 °C by adding Ulp1 Protease at a 1:100 ratio. CoaBC was concentrated and loaded on a Superdex 200 column equilibrated with 25 mM TRIS pH 8.0, 150 mM NaCl. Fraction purity was determined by SDS-page and the purest fractions were pooled, concentrated to ~10 mg mL⁻¹ for *Mtb*CoaBC and 30 mg mL⁻¹ for *Msm*CoaBC, flash frozen in liquid nitrogen and stored at −80 °C.

*E. coli* BL21(DE3) containing the *M. smegmatis* CoaB construct with a *N*-terminal non-cleavable 6xHis tag was grown and harvested as above and re-suspended in 20 mM HEPES pH 7.0, 500 mM NaCl, 20 mM imidazole, 5 mM $MgCl_2$ with protease inhibitors tablets (Roche) and DNAseI (Sigma). Cells were lysed with an Emulsiflex (Avestin) and cell lysate was centrifuged at 27,000× *g* for 30 mins to remove cell debris. Recombinant *M. smegmatis* CoaB was purified with a HiTrap IMAC Sepharose FF column (GE-Healthcare), equilibrated with 20 mM HEPES pH 7.0, 500 mM NaCl and 20 mM imidazole. Elution was carried out in the same buffer with 500 mM imidazole. Protein was concentrated and loaded on a Superdex 200 column equilibrated with 20 mM HEPES pH 7.0 and 500 mM NaCl. Fraction purity was assessed by SDS-page and the purest fractions were pooled concentrated to 22 mg mL⁻¹, flash frozen in liquid nitrogen and stored at −80 °C.

*E. coli* BL21(DE3) containing a human CoaB construct with a cleavable *N*-terminal 6xHis tag was grown and harvested as above and re-suspended in 50 mM TRIS pH 8.0, 250 mM NaCl, 20 mM imidazole, 5 mM $MgCl_2$ with protease

inhibitors tablets (Roche) and DNAseI (Sigma). Cells were lysed using a sonicator and cell lysate was centrifuged at 27,000x g for 30 mins to remove cell debris. Recombinant Human CoaB was purified with a HiTrap IMAC Sepharose FF column (GE-Healthcare), equilibrated with 50 mM TRIS pH 8.0, 250 mM NaCl and 20 mM imidazole. Elution was performed in the same buffer with 300 mM Imidazole. Protein was dialysed in 25 mM TRIS pH 8 and 150 mM NaCl and the His tag was cleaved overnight at 4 °C by adding TEV Protease at a 1:50 ratio. Human CoaB was concentrated and loaded on a Superdex 200 column equilibrated with 25 mM TRIS pH 8.0, 150 mM NaCl. Fraction purity was determined by SDS-page and the purest fractions were pooled, concentrated to ~10 mg mL$^{-1}$, flash frozen in liquid nitrogen and stored at −80 °C. SDS-PAGE images showing the pure enzymes are given in (Supplementary Fig. 11).

**Native mass spectrometry.** Spectra were recorded on a Synapt HDMS mass spectrometer (Waters) modified for studying high masses. *Mtb*CoaBC and *Msm*CoaBC were exchanged into NH$_4$OAc (500 mM, pH 7.0) solution using Micro Bio-Spin 6 chromatography columns (Bio-Rad). A sample volume of 2.5 μL was injected into a borosilicate emitter (Thermo Scientific) for sampling. Instrument conditions were optimised to enhance ion desolvation while minimising dissociation of macromolecular complexes. Typical conditions were capillary voltage 1.8–2.0 kV, sample cone voltage 100 V, extractor cone voltage 1 V, trap collision voltage 60 V, transfer collision voltage 60 V, source temperature 20 °C, backing pressure 5 mbar, trap pressure 3–4 × 10$^{-2}$ mbar, IMS (N$_2$) pressure 5–6 × 10$^{-1}$ mbar and TOF pressure 7–8 × 10$^{-7}$ mbar. Spectra were calibrated externally using caesium iodide. Data acquisition and processing were performed using MassLynx 4.1 (Waters).

**Crystallisation.** For both full-length *Msm*CoaBC and *Msm*CoaB alone, the crystallisation screens and optimisation were performed at 18 °C using the sitting-drop vapour diffusion method. For CoaBC 300 nL of pure protein at 30 mg mL$^{-1}$, preincubated with 3 mM CTP and 10 mM MgCl$_2$, was mixed in 1:1 and 1:2 (protein to reservoir) ratio with well solution using a mosquito robot (TTP labtech). Initial conditions were obtained in the Classics lite crystallisation screen (Qiagen), solution 1. Crystals obtained in this condition diffracted poorly, therefore several rounds of optimisation were performed. The final optimised condition consisted of 0.1 M BisTris pH 6.5, 10 mM CoCl$_2$ 0.8 M 1,6-hexanediol. Crystals appeared after three days in both conditions. A cryogenic solution was prepared by adding ethylene glycol up to 30% (v/v) to the mother liquor. Crystals were briefly transferred to this solution, flash frozen in liquid nitrogen and stored for data collection.

For *Msm*CoaB, 200 nL of pure protein at 22–24 mg mL$^{-1}$ with 10 mM CTP was mixed in 1:1 ratio with well solution using a Phoenix robot (Art Robbins). Crystals were obtained in Wizards classics III&IV (Rigaku) solution G4 consisting of 20% (w/v) PEG 8000, 0.1 M MES pH 6.0 and 0.2 M calcium acetate. Crystals appeared after 2 days.

To obtain ligand-bound structures, soaking was performed using the hanging-drop vapour diffusion method as follows: 2 μL of a solution containing 20% (w/v) PEG 8000, 0.1 M MES pH 6.0, 0.2 M calcium acetate, 0.25 M NaCl 10% (v/v) DMSO and 1–5 mM inhibitors was left to equilibrate against 500 μL of reservoir solution for 3 days. Crystals were then transferred to the pre-equilibrated drops and incubated for 24 h. A cryogenic solution was prepared by adding 2-methyl-2,4-pentanediol up to 25% (v/v) to mother liquor. Crystals were briefly transferred to this solution, flash frozen in liquid nitrogen and stored for data collection.

**Data collection and processing.** The data sets were collected at stations I02 and I03 at Diamond Light Source (Oxford, UK). The diffraction images were processed with AutoPROC[47] using XDS[48] for indexing and integration with AIMLESS[49] and TRUNCATE[50] from CCP4 Suite[51] for data reduction, scaling and calculation of structure factor amplitudes and intensity statistics.

**Structure solution and refinement.** *Msm*CoaB and *Msm*CoaBC structures were solved by molecular replacement using PHASER[52] from the PHENIX software package[53]. For *Msm*CoaB, the atomic coordinates of *Msm*CoaB structure (PDB: 4QJI) were used as a search model. Ligand-bound structures were solved using our highest resolution *Msm*CoaB apo form structure (PDB: 6TH2). For *Msm*CoaBC, atomic coordinates of *Arabidopsis thaliana* CoaC (PDB: 1MVL)[22] and our highest resolution CoaB structure (PDB: 6TH2) were used as search models. Model building was done with Coot[54] and refinement was performed in PHENIX[53]. Structure validation was performed using Coot and PHENIX tools[53,54]. All figures were prepared using Pymol (The PyMOL Molecular Graphics System, Version 2.0 Schrödinger, LLC.) and ligand interactions calculated with Arpeggio[55].

**High-throughput screening.** Potential inhibitors of *Mtb*CoaBC were assessed at room temperature using a PHERAstar microplate reader (BMG Labtech). Pyrophosphate produced by CoaB was converted to two molecules of inorganic phosphate using a pyrophosphatase. Phosphate was then detected using the BIOMOL® Green reagent (Enzo Life Sciences), which when bound to phosphate absorbs light at 650 nm. An end-point assay was carried out in clear, flat-bottom, polystyrene, 384-well plates (Greiner) in an 50 μL reaction volume containing

100 mM TRIS, pH 7.6, 1 mM MgCl$_2$, 1 mM TCEP, 0.03 U/mL pyrophosphatase, 2 μM CTP, 40 μM L-cysteine, 30 μM PPA and 30 nM *Mtb*CoaBC. Assays were performed by adding 25 μL of a 2-times concentrated reaction mixture containing all components with the exception of the enzymes to all wells, and the reactions started by adding 25 μL of a 2-times concentrated enzyme mixture. The reaction was carried out for 2 h at room temperature, before 50 μL of BIOMOL® Green reagent was added and incubated for a further 20 min prior to reading.

**Inorganic pyrophosphatase-purine nucleoside phosphorylase PNP-PPIase assay.** The commercially available EnzChek pyrophosphate assay kit (E-6645) (Life Technologies) was used for this assay. The final reaction composition used was 0.03 U/mL inorganic pyrophosphatase, 1 U/mL purine nucleoside phosphorylase, 1 mM MgCl$_2$, 200 μM MESG, 100 mM TRIS pH 7.5, 1 mM TCEP, 2% DMSO, 32 nM *Mtb*CoaBC or *Msm*CoaBC, 125 μM CTP, 125 μM PPA, 500 μM L-cysteine, and various concentrations of compounds being tested for inhibition, all prepared from DMSO stock solutions (compounds of series one and two) or water (CoA and CoA thioesters). In the assays with HCoaB an enzyme concentration of 1 μM was used instead. Controls were performed to assess the activity of all the compounds tested in this work against the two coupling enzymes and the compounds showed an absence of inhibition at the tested range of concentrations. Assays were performed on either a CLARIOstar or PHERAstar microplate reader (BMG Labtech) in 96-well plates (Greiner). A substrate mixture containing the substrates and the inhibitor was pre-incubated at 25 °C for 10 min. An enzyme solution was prepared and separately pre-incubated at 25 °C for 10 min. The reaction was initiated by the addition of the substrates to the solution containing the enzyme to a final volume of 75 μL. Enzymatic activity was monitored by following the absorbance at 360 nm for 30 min (100 cycles/20 s each cycle) and only the steady-state region of the assay was used for data analysis. Assays were performed in triplicates, including a negative control (lacking PPA) and a positive control (lacking inhibitor).

Competition assays to assess the first half of the CoaB enzymatic reaction were performed using the same conditions but with variable substrate concentrations (31.25, 62.5, 125, 250 and 500 μM for CTP and PPA.

**CMP quantification assay.** To assess the second half of the CoaB enzymatic reaction the release of CMP was monitored using an I-Class UPLC (waters). The 75 μL enzymatic reactions contained 1 mM MgCl$_2$, 100 mM TRIS pH 7.5, 1 mM TCEP, 2% DMSO, 32 nM *Mtb*CoaBC, 125 μM CTP, 125 μM PPA, variable concentrations of L-cysteine (31.25 μM, 62.5 μM, 93.75 μM 125 μM and 250 μM), and various concentrations of compounds 1b, 2b (prepared from DMSO stocks) and acetyl-CoA. Reactions were incubated at 25 °C and stopped by adding 5 μL of EDTA to a final concentration of 20 mM at 5 min intervals for a period of 15 min. 120 μL of mobile phase solvent A (0.1% formic acid in water) was added to each of the reactions and 10 μL of every reaction were injected into an Acquity UPLC HSS T3 column, 2.1 mm diameter, 150 mm length and with a particle size of 1.8 μm (Waters) and eluted using a gradient elution consisting of Mobile Phase A: 0.1% formic acid in water and mobile phase B: 0.1% formic acid in 100% acetonitrile for 4 min. The absorbance was monitored at 280 nm using a photodiode array (PDA) detector (Waters). All reactions were carried out in triplicate.

**M. tuberculosis strains and MIC measurements.** MIC determination for *M. tuberculosis* H37RvMA was performed as previously described[56] in the following media: 7H9/ADC/glycerol (4.7 g/L Difco Middlebrook 7H9 base, 100 mL/L Middlebrook albumin (BSA)-dextrose-catalase (ADC) Difco Middlebrook, 0.2% glycerol and 0.05% Tween-80), 7H9/Cholesterol/Tyloxapol (4.7 g/L 7H9 base, 0.81 g/L NaCl, 24 mg/L cholesterol, 5 g/L BSA fraction V and 0.05% Tyloxapol) and GAST/Fe (0.3 g/L of Bacto Casitone (Difco), 4.0 g/L of dibasic potassium phosphate, 2.0 g/L of citric acid, 1.0 g/L of L-alanine, 1.2 g/L of magnesium chloride hexahydrate, 0.6 g/L of potassium sulfate, 2.0 g/L of ammonium chloride, 1.80 ml/L of 10 N sodium hydroxide, 10.0 mL of glycerol 0.05% Tween-80 and 0.05 g of ferric ammonium citrate adjusted to pH 6.6).

**M. tuberculosis CoaBC knockdown and on-target activity.** *M. tuberculosis* H37RvMA and its *coaBC* Tet-OFF conditional knockdown derivative[12] were grown in Difco Middlebrook 7H9 broth (BD) supplemented with Middlebrook albumin-dextrose-catalase (ADC) enrichment (BD), 0.2% glycerol (Sigma-Aldrich) and 0.05% Tween-80, unless otherwise indicated. Hygromycin and kanamycin were used at final concentrations of 50 μg/mL and 25 μg/mL, respectively, and pantethine (Sigma-Aldrich) supplementation was included at 2.5 mg/mL where required. The anhydrotetracycline (ATc) inducer was used at concentrations up to 200 ng/mL in order to transcriptionally silence *coaBC* in the Tet-OFF hypomorph.

The minimum inhibitory concentrations (MIC$_{99}$) of the compounds were determined by measuring fluorescence output using Alamar Blue, as previously described (Singh et al.[56]). Briefly, 2-fold serial dilutions of compound were inoculated with a suspension of *M. tuberculosis* at a cell density of ~10$^5$ CFU/mL in a 96-well microtiter plate and incubated at 37 °C for 10 days, following which 10 μL Alamar Blue solution was added and the plates were incubated for a further 24 h. Fluorescence as an indication of growth was measured using a SpectraMax i3x

Multi-Mode Microplate Reader (Molecular Devices) in bottom-reading mode with excitation at 544 nm and emission at 590 nm.

**Mycobacterium tuberculosis compound uptake.** *M. tuberculosis* H37Rv was grown to OD$_{580}$ 1 in Middlebrook 7H9 supplemented with 0.2% glucose and 0.2% glycerol. One mL was added to each nylon Durapore 0.22 μm membrane filter. Bacteria-laden filters were placed atop Middlebrook 7H11 agar supplemented with 0.2% glycerol and 10% OADC, incubated at 37 °C for 1 week to expand the biomass, and transferred to lay atop a reservoir containing 7H9 medium with 0.2% glucose and 0.2% glycerol (no tyloxapol) and 10 μM of test agent. After 24 h at 37 °C, bacteria-laden filters were plunged into 1 mL 40:40:20 methanol:acetonitrile:water pre-cooled to −20 °C and lysed in a bead beater while spent media was reserved for extraction and analysis with cell lysates. Both cell lysates and spent media were mixed with equal volumes of 50% acetonitrile, 0.2% formic acid for mass spectrometry in positive and negative ion mode as previously described[57].

**Reporting summary**. Further information on research design is available in the Nature Research Reporting Summary linked to this article.

## Data availability
Coordinates and structure factors related to this work have been deposited in the PDB with accession numbers: 6TGV, 6TH2 and 6THC. The source data for Tables 1, 2 and Figs. 4, 6 and Supplementary Fig. 7 are provided as a source data file. Other data are available from the corresponding authors upon reasonable request. Source data are provided with this paper.

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

## Acknowledgements

We dedicate this work to the memory of Professor Chris Abell who sadly died suddenly on Monday 26th October at the age of 62. Chris developed highly interdisciplinary research focused on understanding the mechanisms of key enzymes and the development of chemistry for structure-guided fragment-based drug discovery. He leaves a lasting legacy through his work and the numerous scientists whose careers he helped to shape. This work was funded by the Bill and Melinda Gates Foundation HIT-TB (OPP1024021) and SHORTEN-TB (OPP1158806) (VMendes, JCE and MB) and in part by the Intra-mural Research Program of NIH, NIAID (HIMB and CEB) and the South African Medical Research Council and National Research Foundation (VMizrahi). J.H. was financially supported by Swiss National Science Foundation (SNSF Early PostDoc. Mobility fellowship, P2ZHP2_164947) and the Marie Curie Research Grants Scheme, EU H2020 Framework Programme (H2020-MSCA-IF-2017, ID: 789607). C.S. was funded in part by a NHMRC Overseas Biomedical Fellowship (1016357) and in part by the Bill and Melinda Gates Foundation HIT-TB (OPP1024021). CoaBC screening was funded by a MRC-CinC (grant no. MC_PC_14099). T.L.B. is funded by the Wellcome Trust (Wellcome Trust Investigator Award 200814_Z_16_Z: RG83114). The authors would like to thank the Diamond Light Source for beam-time (proposals mx9537, mx14043, mx18548), the Seattle Structural Genomics Centre for Infectious Disease for kindly providing the *M. smegmatis* CoaB plasmid and Dr. Nuno Empadinhas for graciously providing the DNA of *M. smegmatis* mc$^2$ 155.

## Author contributions

V.Mendes wrote the manuscript. V.Mendes designed and performed all the crystal-lographic experiments with the help of M.B., O.B. and J.C.W. V.Mendes and J.H. designed and performed the kinetic experiments. J.H. synthesised 4′-phosphopantothe-nate. P.H.M.T. performed docking experiments. D.S.C. performed the native mass spectrometry experiments. S.G., T.B., S.O'N., S.D., J.P. and C.S. developed and performed the high-throughput screening. J.C.E., S.L.L. and H.I.M.B. performed the microbiology experiments on *M. tuberculosis* H37Rv. Z.W. and N.N. performed the compound uptake experiments. V.Mendes, J.C.E., S.G., A.G.C., P.C.R., K.Y.R., C.A., H.I.M.B., C.E.B., V.Mizrahi, P.G.W. and T.L.B. managed the project. All authors approved the manuscript.

## Competing interests

The authors declare no competing interests.
