## [Peer Review File · Nature Communications]

Reviewers' comments:

Reviewer #1 (Remarks to the Author):

Bifunctional CoaBC catalyzes within the coenzyme A biosynthetic pathway the synthesis of 4'-phosphopantetheine from 4'-phosphopantothenate and cysteine using CTP and FMN as cofactors. The mechanisms of CoaB and CoaC enzymes are well investigated and also their crystal structures (and those of CoaC related dodecameric proteins such as EpiD and MrsD) have already been determined more than 15 years ago. In this very well written manuscript by Mendes et al. for the first time the three-dimensional structure of a dodecameric bifunctional CoaBC protein (from *Mycobacterium smegmatis*) has been solved. Mendes et al. also studied the regulation of CoaBC (in this case from *M. tuberculosis*) by CoaA thioesters and identified inhibitors of *M. tuberculosis* CoaB. They used the known crystal structure of *M. smegmatis* CoaB to solve the crystal structure of CoaB with bound inhibitor and found an allosteric site.

(1) The dodecameric structure of CoaBC was confirmed in an elegant way by native mass spectrometry. However, the experimentally determined masses were 1-2% higher than calculated. Did the authors include the mass of 12 X FMN (tightly bound cofactor) into the calculation of the theoretical mass?

(2) As important it is to have the crystal structure of dodecameric CoaBC with bound cofactors CTP and FMN there is not so much new biochemistry we can learn from it. The importance of a long loop in substrate channeling from the CoaB to the CoaC active site is only discussed and structures of CoaBC with bound substrates, reaction intermediates or reaction products have not been solved.

(3) A model for a bifunctional dodecameric CoaBC protein has been published before by Manoj et al. (Structure, 2003).

(4) The regulation of *M. tuberculosis* CoaBC by Coenzyme A and CoA thioesters is an interesting novel finding and the authors suggest that acyl-CoAs block the active-site (but this proposed mode of acyl-CoA binding was not confirmed by a crystal structure).

(5) The major part of the paper is about identifying inhibitors of the *M. tuberculosis* CoaBC enzyme to use them as novel antibiotics combatting *M. tuberculosis*. Every approach to combat *M. tuberculosis* and thus tuberculosis is of course very welcome and of great importance. However, I would recommend a more cautious discussion if coenzyme A biosynthesis is really a good target, even if it looks like, that in *M. tuberculosis* CoaBC deficiency cannot be rescued by pantetheine uptake (Evans et al. 2016). There is still the general problem, that the human coenzyme A biosynthesis machinery is similar to the prokaryotic one.

(6) Therefore, it should be tested if enzyme inhibitors are really specific for *M. tuberculosis* enzymes and do not inhibit the human enzymes. For the new CoaB inhibitors identified in the present study this has not been done. Moreover, these new compounds cannot be used as antibiotics since they are not taken up by *M. tuberculosis*. However, they still may serve as new and important lead structures for the design of novel antibiotics.

(7) For this, it is crucial to elucidate their exact mode of action. To do so, Mendes et al. performed kinetic studies using *M. tuberculosis* CoaBC and solved the crystal structure of *M. smegmatis* CoaB with bound inhibitor. It would be more convincing if related studies were done with enzymes originated from the same organism. Why was it not feasible to solve the crystal structure of *M. tuberculosis* CoaB with bound inhibitor or to repeat the kinetic studies with *M. smegmatis* CoaBC and to solve the crystal structure of *M. smegmatis* CoaBC with bound inhibitors (considering that CoaB in solution is not forming dimers and therefore is not active)?

(8) The authors used the release of PPI to quantify the CoaB activity of the CoaBC enzyme and to

study the new inhibitors. This can be done, however important information may be lost quantifying just PPI. CoaB synthesizes 4'-phosphopantothenoylcysteine by a) formation of 4'-phosphopantothenoyl-CMP (is this released by the enzyme if the second half reaction is blocked?; and PPI) from 4'-phosphopantothenate and CTP and b) conversion of 4'-phosphopantothenoyl-CMP by reaction with cysteine. In principle, the authors only analyzed reaction a) (but included cysteine into their kinetic analysis). In order to analyze the mode of inhibition in more detail (accumulation of 4'-phosphopantothenoyl-CMP?, compare point 9) the formation of 4'-phosphopantothenoyl-CMP and 4'-phosphopantothenoylcysteine should also be analyzed and quantified by for example HPLC-MS. For the interpretation / discussion of the kinetic and the structural data it is important, that binding of cysteine only occurs after formation of the 4'-phosphopantothenoyl-CMP intermediate (Stanitzek et al., Structure 2004).

(9, related to point 8): In figure S3 the allosteric site residues of CoaB are marked. One of these is the strictly conserved Asn residue within the sequence motif ... GNRSSGK. For the E. coli CoaB enzyme it has been shown that exchange of this Asn residue (N210) blocks the second half reaction of CoaB leading to accumulation of 4'-phosphopantothenoyl-CMP (Kupke, J. Biol. Chem. 2002 and Stanitzek et al., Structure 2004). This indicates that N210 is part of the active site and involved in binding cysteine, but is not an allosteric site residue.

There are more residues proposed to be allosteric site residues but belong in the E. coli CoaB structure to the nucleotide binding motif I (Ala275-Ala-Val-Ala-Asp-X9-Lys289-X-Lys-Lys). The authors should comment on this in their manuscript to clarify if the binding site of the inhibitors is overlapping with the active site or if some residues are both, active site and allosteric site residues.

(10) Kumar et al. showed (BBRC, 2007) that 4'-phosphopantothenol inhibits competitively the utilization of 4'-phosphopantothenate by M. tuberculosis CoaBC. This information should be added either to the introduction or the discussion section (and compared with the proposed allosteric mechanism of the newly identified CoaB inhibitors).

(11) The authors should add to the supplement figures of SDS-gels showing purification of CoaBC and CoaB proteins used for crystal structure analysis and enzyme kinetics.

(12) The "Competing Interests" declaration is missing.

Reviewer #2 (Remarks to the Author):

In this manuscript, Mendes and collaborators report the structural and biochemical characterization of an essential enzyme involved in CoA synthesis. The CoaAB enzyme is essential for mycobacterial survival and as such represents a very attractive drug target. Thanks to a high-throughput drug screening assay, the authors identified two promising compounds inhibiting the enzyme. The crystal structure of one of these compounds bound to CoaB demonstrates that it sits in an allosteric site.

Overall this study is of great interest, as first of all, this is the first crystal structure reported of a full-length CoaAB enzyme. The structural analysis supported by excellent biochemical approaches brings new knowledge into the catalysis mediated by CoaAB.

The fact that authors discovered potent allosteric inhibitors may also pave the way to new molecules inhibiting Mycobacterium tuberculosis growth, which is urgently needed.

Overall the data are sound, the crystallographic, biochemical and enzymatic approaches are very well conducted.

The main concern of this article is regarding the MIC determination and M. tuberculosis growth inhibition. The authors claim that all compounds tested have moderate activity against M. tuberculosis H37Rv growth, which does not correlate with the pretty good activity of the inhibitors in vitro against CoaAB. In the discussion, the authors propose that this low activity could be due to

general issues encountered when designing inhibitors against mycobacteria i.e. low permeability, efflux pumps, etc...

One aspect that the authors did not mention is that CoaB might not be the target of these inhibitors in vivo. Unfortunately, this was not tested in this study.

To improve this manuscript and particularly the major aspect of drug target :

1-The authors should assess if CoaB overexpression in *M. tuberculosis* H37Rv strain is increasing the MIC of the "best" inhibitors i.e. compounds 1b, 2b, 2c. This could be achieved by transforming *M. tuberculosis* H37Rv strain with a high copy plasmid allowing expression of CoaB or a plasmid expressing CoaB under the control of a strong promoter. As work with *M. tuberculosis* can be lengthy due to the slow growth of the strain this could also be achieved instead with *M. smegmatis* if compounds are of course active against this strain.

Minor points :

-line 349: the authors mention "extremely" low sequence identity, low sequence identity would be more appropriate for 25% identity.

-line 404. replace "modest vitro activity" by "modest in vitro activity"

-In the two crystal structures of CoaB the authors placed a Calcium ion instead of a Magnesium as seen in CoaAB structure.

The distances between the ions, phosphate groups and residues in the vicinity are however identical in the 3 crystal structures which might indicate that it is the same ion in the 3 structures. Did the authors calculated an anomalous map to assess the presence of Calcium ?. The wavelength at which data were collected might not be optimal for Calcium but it might be sufficient to see an anomalous peak and distinguish between the two ions.

Reviewer #3 (Remarks to the Author):

The current manuscript reported the first structure of a full-length MsmCoaBC, and identified two CoaB inhibitors with low anti *M. tuberculosis* effects. The manuscript is well written and is of interests for the researchers in this field. However, to meet the standard of Nature Communication, the following major revision needs to be done.

In the introduction, it is difficult to follow the biosynthesis of CoA, and a figure should be given. Please indicate the roles of CoaB and CoaC, and the difference of individual CoaB and CoaC in combination and the CoaBC.

In the introduction, the known relative inhibitors of CoaB should be introduced if there are any.

The PDB ID of crystal structures of CoaB and CoaC should be given. As the authors mentioned that the differences of CoaB component in MsmCoaBC and individual MsmCoaB; and CoaC in MsmCoaC and individual MsmCoaC are small, are there any previous studies showed the protein-protein interactions of the individual MsmCoaB and the individual MsmCoaC ? Please indicate why such interactions are important. In another word, why it is important to get the crystal structure of MsmCoaBC.

The structures of the compounds 1b and 2b identified cannot be structurally specific to the MsmCoaB. Especially compound 2b is a flavone compound, which should have multiple biological targets. Can authors explain why such kind of compounds was chosen?

Structure activity relationship should be summarized. For my personal view, the catechol group looks very important. The chemical structures of important compounds should be list in the main text rather in the supporting information.

Can authors explain that why compound 1b and 2b showed different mechanism?

“Compound 2b shows mixed inhibition relative to CTP with a K_i of 0.093 μM and uncompetitive inhibition for PPA and L-cysteine and PPA with a αK_i respectively of 0.062 and 0.049 μM (Figure 4C and Table 2).” Can the authors check and rewords this sentence?

For the inhibitor discovery, why there is no positive control in the assays?

If the allosteric inhibitors have influences in the biosynthetic pathway of CoA in cellular level? If it is possible, I would like to suggest the authors considering some assays for that to confirm if 1b or 2b have such influences.

Is there any link between the binding mode study of compound 1b and crystal structure of CoaBC? It looks like that the manuscript is composed by two separate parts, which is the crystallization of CoaBC and the inhibitor identification (does not based on the crystallization of CoaBC). I would like to suggest the author explain the link between them.

Hao Wang

Reviewers' comments:

Reviewer #1 (Remarks to the Author):

Bifunctional CoaBC catalyzes within the coenzyme A biosynthetic pathway the synthesis of 4'-phosphopantetheine from 4'-phosphopantothenate and cysteine using CTP and FMN as cofactors. The mechanisms of CoaB and CoaC enzymes are well investigated and also their crystal structures (and those of CoaC related dodecameric proteins such as EpiD and MrsD) have already been determined more than 15 years ago. In this very well written manuscript by Mendes et al. for the first time the three-dimensional structure of a dodecameric bifunctional CoaBC protein (from *Mycobacterium smegmatis*) has been solved. Mendes et al. also studied the regulation of CoaBC (in this case from *M. tuberculosis*) by CoaA thioesters and identified inhibitors of *M. tuberculosis* CoaB. They used the known crystal structure of *M. smegmatis* CoaB to solve the crystal structure of CoaB with bound inhibitor and found an allosteric site.

(1) The dodecameric structure of CoaBC was confirmed in an elegant way by native mass spectrometry. However, the experimentally determined masses were 1-2% higher than calculated. Did the authors include the mass of 12 X FMN (tightly bound cofactor) into the calculation of the theoretical mass?

We have mistakenly not included the FMN mass in the calculation of the theoretical mass. This has now been corrected. Page 5 Lines 121-123.

(2) As important it is to have the crystal structure of dodecameric CoaBC with bound cofactors CTP and FMN there is not so much new biochemistry we can learn from it. The importance of a long loop in substrate channeling from the CoaB to the CoaC active site is only discussed and structures of CoaBC with bound substrates, reaction intermediates or reaction products have not been solved.

While the authors agree with the reviewer that these structures would be nice to have, many attempts to obtain structures with all the substrates and reaction intermediates were performed using different crystallization methods, several mycobacterial species and protein constructs but crystals did not diffract sufficiently to solve the structures.

(3) A model for a bifunctional dodecameric CoaBC protein has been published before by Manoj et al. (Structure, 2003).

Manoj and colleagues proposed a theoretical model for the bacterial CoaBC based on the Human CoaB and the CoaC homologue EpiD. While the model was a good approximation to reality and allowed to see how a bacterial CoaBC structure would look like it was just a theoretical model with limited usefulness. Therefore we do not think this manuscript should be referenced for this particular situation although we refer it for other reasons.

(4) The regulation of *M. tuberculosis* CoaBC by Coenzyme A and CoA thioesters is an interesting novel finding and the authors suggest that acyl-CoAs block the active-site (but this proposed mode of acyl-CoA binding was not confirmed by a crystal structure). As above, we have performed many attempts to obtain these structures, but again the resolution of the obtained crystals was not enough to model confidently the acyl-CoAs, even though we could observe extra electron density occupying the substrate sites. Nevertheless the experiments we performed clearly show competitive inhibition of CoaBC by acetyl-CoA in relation to PPA and CTP, supporting that the acyl-CoAs indeed bind to the active site.

(5) The major part of the paper is about identifying inhibitors of the *M. tuberculosis* CoaBC enzyme to use them as novel antibiotics combatting *M. tuberculosis*. Every

approach to combat *M. tuberculosis* and thus tuberculosis is of course very welcome and of great importance. However, I would recommend a more cautious discussion if coenzyme A biosynthesis is really a good target, even if it looks like, that in *M. tuberculosis* CoaBC deficiency cannot be rescued by pantetheine uptake (Evans et al. 2016). There is still the general problem, that the human coenzyme A biosynthesis machinery is similar to the prokaryotic one.

While there is some similarity between Human and MtbCoaB (~25%) we now show that the compounds are ~400-800 fold more active for mycobacterial CoaB than for the human orthologue, demonstrating that there is scope to develop selective inhibitors for mycobacterial CoaB targeting this site. Page 10, lines 245-250. We further show that compound 1b is not taken up by Mtb cells and its activity is off target while compound 2b suffer metabolism and again the observed activity is off target. Page 15 lines 351-363.

(6) Therefore, it should be tested if enzyme inhibitors are really specific for *M. tuberculosis* enzymes and do not inhibit the human enzymes. For the new CoaB inhibitors identified in the present study this has not been done. Moreover, these new compounds cannot be used as antibiotics since they are not taken up by *M. tuberculosis*. However, they still may serve as new and important lead structures for the design of novel antibiotics.

See answer above.

(7) For this, it is crucial to elucidate their exact mode of action. To do so, Mendes et al. performed kinetic studies using *M. tuberculosis* CoaBC and solved the crystal structure of *M. smegmatis* CoaB with bound inhibitor. It would be more convincing if related studies were done with enzymes originated from the same organism. Why was it not feasible to solve the crystal structure of *M. tuberculosis* CoaB with bound inhibitor or to repeat the kinetic studies with *M. smegmatis* CoaBC and to solve the crystal structure of *M. smegmatis* CoaBC with bound inhibitors (considering that CoaB in solution is not forming dimers and therefore is not active)?

We now show that the best compound of each series have similar potency for both Mtb and Msm enzymes. Page 10 lines 245-248 We have attempted to obtain crystals Msm and Mtb full length CoaBC with the inhibitors but these never diffracted sufficiently to solve the structures. And in the case of MtbCoaBC even the apo and CTP bound forms did not diffract sufficiently. We have also tried to crystallise MtbCoaB but without success.

(8) The authors used the release of PPI to quantify the CoaB activity of the CoaBC enzyme and to study the new inhibitors. This can be done, however important information may be lost quantifying just PPI. CoaB synthesizes 4'-phosphopantothencysteine by a) formation of 4'-phosphopantothencyl-CMP (is this released by the enzyme if the second half reaction is blocked?; and PPI) from 4'-phosphopantothenate and CTP and b) conversion of 4'-phosphopantothencyl-CMP by reaction with cysteine. In principle, the authors only analyzed reaction a) (but included cysteine into their kinetic analysis). In order to analyze the mode of inhibition in more detail (accumulation of 4'-phosphopantothencyl-CMP?, compare point 9) the formation of 4'-phosphopantothencyl-CMP and 4'-phosphopantothencysteine should also be analyzed and quantified by for example HPLC-MS. For the interpretation / discussion of the kinetic and the structural data it is important, that binding of cysteine only occurs after formation of the 4'-phosphopantothencyl-CMP intermediate (Stanitzek et al., Structure 2004).

We thank the reviewer for this insightful comment. The reviewer is correct in saying that by quantifying PPI we were only looking directly at the first half of the reaction. We have now performed a new analysis measuring the release of CMP by HPLC for L-cysteine

competition assays. The new results are included in Page 9 line 213, Page 11, lines 254-8, Table 2 and Figures 4B and 6B. However, the type of inhibition remains the same. We have also removed the data that used PPI quantification for L-cysteine competition experiments from the manuscript.

(9, related to point 8): In figure S3 the allosteric site residues of CoaB are marked. One of these is the strictly conserved Asn residue within the sequence motif ... GNRSSGK. For the E. coli CoaB enzyme it has been shown that exchange of this Asn residue (N210) blocks the second half reaction of CoaB leading to accumulation of 4'-phosphopantothenoyl-CMP (Kupke, J. Biol. Chem. 2002 and Stanitzek et al., Structure 2004). This indicates that N210 is part of the active site and involved in binding cysteine, but is not an allosteric site residue.

There are more residues proposed to be allosteric site residues but belong in the E. coli CoaB structure to the nucleotide binding motif I (Ala275-Ala-Val-Ala-Asp-X9-Lys289-X-Lys-Lys). The authors should comment on this in their manuscript to clarify if the binding site of the inhibitors is overlapping with the active site or if some residues are both, active site and allosteric site residues.

Several allosteric site residues are shared with the active site: R207, N210, A280, D281, I292, K293 and K294. This has now been included in the text page 12 lines 285 and 286

"There are more residues proposed to be allosteric site residues but belong in the E. coli CoaB structure to the nucleotide binding motif I (Ala275-Ala-Val-Ala-Asp-X9-Lys289-X-Lys-Lys)".

In figure S3 these residues are shaded in red and are part of the CoaB dimer interface not the allosteric site. This is stated in the figure legend.

(10) Kumar et al. showed (BBRC, 2007) that 4'-phosphopantothenol inhibits competitively the utilization of 4'-phosphopantothenate by M. tuberculosis CoaBC. This information should be added either to the introduction or the discussion section (and compared with the proposed allosteric mechanism of the newly identified CoaB inhibitors).

We have added this information to the introduction page 4 lines 91-92. However we feel that discussing a substrate-mimicking inhibitor is tangential to the main topic, the finding of an allosteric site and allosteric inhibitors.

(11) The authors should add to the supplement figures of SDS-gels showing purification of CoaBC and CoaB proteins used for crystal structure analysis and enzyme kinetics. This is now available (figure S11).

(12) The "Competing Interests" declaration is missing. This is now available.

Reviewer #2 (Remarks to the Author):

In this manuscript, Mendes and collaborators report the structural and biochemical characterization of an essential enzyme involved in CoA synthesis. The CoaAB enzyme is essential for mycobacterial survival and as such represents a very attractive drug target. Thanks to a high-throughput drug screening assay, the authors identified two promising compounds inhibiting the enzyme. The crystal structure of one of these compounds bound to CoaB demonstrates that it sits in an allosteric site.

Overall this study is of great interest, as first of all, this is the first crystal structure

reported of a full-length CoaAB enzyme. The structural analysis supported by excellent biochemical approaches brings new knowledge into the catalysis mediated by CoaAB. The fact that authors discovered potent allosteric inhibitors may also pave the way to new molecules inhibiting Mycobacterium tuberculosis growth, which is urgently needed. Overall the data are sound, the crystallographic, biochemical and enzymatic approaches are very well conducted.

The main concern of this article is regarding the MIC determination and M. tuberculosis growth inhibition. The authors claim that all compounds tested have moderate activity against M. tuberculosis H37Rv growth, which does not correlate with the pretty good activity of the inhibitors in vitro against CoaAB. In the discussion, the authors propose that this low activity could be due to general issues encountered when designing inhibitors against mycobacteria i.e. low permeability, efflux pumps, etc..

One aspect that the authors did not mention is that CoAB might not be the target of these inhibitors in vivo. Unfortunately, this was not tested in this study.

To improve this manuscript and particularly the major aspect of drug target :

1-The authors should assess if CoaAB overexpression in M. tuberculosis H37Rv strain is increasing the MIC of the "best" inhibitors i.e. compounds 1b, 2b, 2c. This could be achieved by transforming

M. tuberculosis H37Rv strain with a high copy plasmid allowing expression of CoaAB or a plasmid expressing CoaAB under the control of a strong promoter. As work with M. tuberculosis can be lengthy due to the slow growth of the strain this could also be achieved instead with M. smegmatis if compounds are of course active against this strain.

We have now performed work using an inducible CoaBC knock-down system in a modified Mtb H37Rv strain, that the compounds do not become more active as CoaBC expression is reduced. This indicates that their effect is off target. We further show using LC/MS that compound 1b does not enter the Mtb cells and 2b suffers metabolism. This is consistent with the off target effects. Page 15 lines 351-363.

Minor points :

-line 349: the authors mention "extremely" low sequence identity, low sequence identity would be more appropriate for 25% identity.

Done

-line 404. replace "modest vitro activity" by "modest in vitro activity"

Done

-In the two crystal structures of CoaB the authors placed a Calcium ion instead of a Magnesium as seen in CoaAB structure.

The distances between the ions, phosphate groups and residues in the vicinity are however identical in the 3 crystal structures which might indicate that it is the same ion in the 3 structures. Did the authors calculated an anomalous map to assess the presence of Calcium ?. The wavelength at which data were collected might not be optimal for Calcium but it might be sufficient to see an anomalous peak and distinguish between the two ions.

Our M. smegmatis CoaB crystallization condition does not contain any magnesium but contains 0.2 M calcium acetate. The Fo-Fc map after refining, when magnesium was used instead of calcium also showed a positive peak pointing to larger atom being present. This positive peak is no longer visible after refining when calcium is present. In the full CoaBC structure calcium is not present in the crystallization condition and we incubated the protein with 10mM Magnesium.

Reviewer #3 (Remarks to the Author):

The current manuscript reported the first structure of a full-length MsmCoaBC, and identified two CoaB inhibitors with low anti *M. tuberculosis* effects. The manuscript is well written and is of interests for the researchers in this field. However, to meet the standard of Nature Communication, the following major revision needs to be done.

1-In the introduction, it is difficult to follow the biosynthesis of CoA, and a figure should be given. Please indicate the roles of CoaB and CoaC, and the difference of individual CoaB and CoaC in combination and the CoaBC.

A new figure (figure 1) was added to the manuscript that clearly shows the full CoA pathway and the role of every enzyme.

2-In the introduction, the known relative inhibitors of CoaB should be introduced if there are any.

Prior to this study there wasn't any known allosteric inhibitors of CoaB. Almost all the reported inhibitors CoaB inhibitors, which are very few in number, are substrate mimicking. The small set that are not related to the substrates were reported by us in reference 13, are weak inhibitors and their mode of binding is unknown. Nevertheless a sentence was added to the introduction to reflect this. Page 4 lines 91-92

3.1-The PDB ID of crystal structures of CoaB and CoaC should be given.

If the reviewer is referring to the available structures of CoaB and CoaC used for molecular replacement the PDB IDs are available in the methods section. If he is referring to the the CoaB and CoaBC structures deposited for this work they are available throughout the text in the results section (page 6 line 130 and 137). By lapse we have not included the PDB code for the structure with compound 2b this is now in page 12 line 279).

3.2-As the authors mentioned that the differences of CoaB component in MsmCoaBC and individual MsmCoaB; and CoaC in MsmCoaC and individual MsmCoaC are small, are there any previous studies showed the protein-protein interactions of the individual MsmCoaB and the individual MsmCoaC ? Please indicate why such interactions are important. In another word, why it is important to get the crystal structure of MsmCoaBC.

In Mycobacteria (and other bacteria) as CoaB and CoaC occur fused and no protein-protein interactions studies were performed with the individual domains. We also mention that the CoaB domain when expressed individually is inactive and does not form a dimer in solution but we have not worked with the individual CoaC domain. There are studies performed in yeast showing protein-protein interactions between individual CoaB (CAB2) and CoaC (CAB3) (references 41 and 42). However both yeast proteins contain extra regions that account for some of the interactions.

The fact that CoaB alone did not dimerise in solution and was inactive was concerning especially when *E. coli* CoaB dimerises in solution and is active. From a structure biology and drug discovery perspective it was important to verify if our CoaB construct was a good representation of the CoaB domain in the full length CoaBC and explain the reasons behind the differences in behaviour between different species. It was also important to obtain insight on why this organization is important. All of this is already stated in the text both in the results section (sections overall structure of CoaBC and structural basis for inhibition of CoaB by allosteric inhibitors) and also in the discussion.

4-The structures of the compounds 1b and 2b identified cannot be structurally specific to the MsmCoaB. Especially compound 2b is a flavone compound, which should have multiple biological targets. Can authors explain why such kind of compounds was chosen?

The authors agree that the compounds, especially series two is indeed promiscuous. However, since the compounds were by far the most potent inhibitors ever seen for CoaB that are not analogues of a substrate we deemed it was important to study them to understand how their were acting on CoaB as they could serve as a basis for further drug discovery work. Following these compounds allowed the identification of a new allosteric site and we have now proved with the Human CoaB data (page 10, lines 245-250, Table 1 and figure S7) that selectivity towards mycobacterial enzymes can be built in this site.

5-Structure activity relationship should be summarized. For my personal view, the catechol group looks very important. The chemical structures of important compounds should be list in the main text rather in the supporting information.

The chemical structures of the initial hits and most potent compounds of each series are given in a new figure 5. SAR is summarized at the bottom of the results section "structural basis for inhibition of CoaB by allosteric inhibitors" (pages 13 and 14 lines 321-339). We have also added an extra sentence in the end of the section to make clear the importance of the catechol for binding. Page 14 lines 339-341.

6-Can authors explain that why compound 1b and 2b showed different mechanism?

Since we did not obtain a structure with compound 2b we can only hypothesize about the reasons behind the different mechanism, furthermore docking wasn't particularly helpful in trying to understand this as 2b fits well within the allosteric site as figure S10 panel D shows. However the mixed inhibition mechanism for CTP correlates well with the fact that we couldn't obtain structures with the all compounds from series 2 that we have tried. Since the compounds from series 2 are more bulky that the ones from series one it may be possible that slight movements of the amino acids at the bottom of the allosteric site are needed to accommodate these compounds that also interfere with the binding of CTP and these movements were blocked in our crystal structure. We have now included this in the discussion. Pages 17 and 18 lines 422-429.

7-"Compound 2b shows mixed inhibition relative to CTP with a K_i of 0.093 μM and uncompetitive inhibition for PPA and L-cysteine and PPA with a αK_i respectively of 0.062 and 0.049 μM (Figure 4C and Table 2)." Can the authors check and rewords this sentence?

Done

8-For the inhibitor discovery, why there is no positive control in the assays?

At the time the screen was performed there wasn't any commercially available CoaB inhibitor.

9-If the allosteric inhibitors have influences in the biosynthetic pathway of CoA in cellular level? If it is possible, I would like to suggest the authors considering some assays for that to confirm if 1b or 2b have such influences.

The compounds do not have a direct influence on the CoA pathway. Compound 1b cannot enter the cells and compound 2b is metabolised. Both do not act on target. We have added new experiments that show this. Page 15 lines 351-363.

10-Is there any link between the binding mode study of compound 1b and crystal structure of CoaBC? It looks like that the manuscript is composed by two separate parts, which is the crystallization of CoaBC and the inhibitor identification (does not based on the crystallization of CoaBC). I would like to suggest the author explain the link between them.

The authors believe that the link is already stated in the manuscript (sections overall structure of CoaBC and structural basis for inhibition of CoaB by allosteric inhibitors) and that explaining this further would be redundant. We also explain it more concisely while answering the question 3.2.

REVIEWERS' COMMENTS

Reviewer #1 (Remarks to the Author):

Thanks to the authors for their careful consideration of my comments. I do not have further concerns.

Reviewer #2 (Remarks to the Author):

In this revised manuscript, Mendes and collaborators nicely addressed the different remarks and comments of the 1st review. It would have been nice though to try overexpression of the CoaBC to address the target issue and it would have been actually probably easier than doing the silencing. Nonetheless the LC/MS experiments showing that compound 1b does not enter the mycobacterial cells and that 2b suffers metabolism are a really nice addition to the study and clearly explain the low MIC of the compounds. The fact that CoaBC inhibitors are in fact off-target does not alter however the impact of this study as the crystal structures described in here are of great interest. Further, the discovery and characterization of CoaBC inhibitors may pave the way to the development of new inhibitors targeting this pathway and thus to new anti-mycobacterial compounds that are highly needed.

For these reasons, I recommend the acceptance of this manuscript.

Mickael Blaise